# What's "up" with vision-language models?
# Investigating their struggle with spatial reasoning

**Amita Kamath**[1]        **Jack Hessel**[2]        **Kai-Wei Chang**[1]

[1] University of California, Los Angeles

[2] Allen Institute for AI

{kamatha, kwchang}@cs.ucla.edu, jackh@allenai.org

## Abstract

Recent vision-language (VL) models are powerful, but can they reliably distinguish "right" from "left"? We curate three new corpora to quantify model comprehension of such basic spatial relations. These tests isolate spatial reasoning more precisely than existing datasets like VQAv2, e.g., our What'sUp benchmark contains sets of photographs varying *only* the spatial relations of objects, keeping their identity fixed (see Figure 1: models must comprehend not only the usual case of a dog under a table, but also, the same dog *on top of* the same table). We evaluate 18 VL models, finding that all perform poorly, e.g., BLIP fine-tuned on VQAv2, which nears human parity on VQAv2, achieves 56% accuracy on our benchmarks vs. humans at 99%. We conclude by studying causes of this surprising behavior, finding: 1) that popular vision-language pre-training corpora like LAION-2B contain little reliable data for learning spatial relationships; and 2) that basic modeling interventions like up-weighting preposition-containing instances or fine-tuning on our corpora are not sufficient to address the challenges our benchmarks pose. We are hopeful that these corpora will facilitate further research, and we release our data and code at https://github.com/amitakamath/whatsup_vlms.

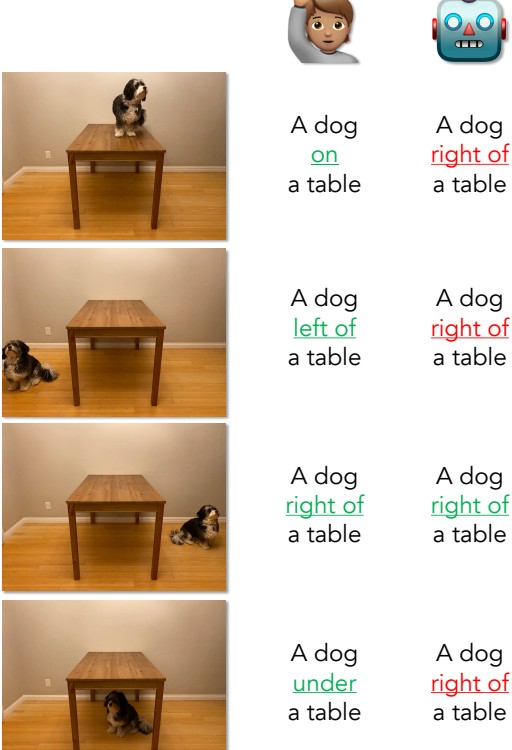

Figure 1: We propose three tightly controlled benchmarks to assess model capacity for fine-grained spatial reasoning, showing that popular vision-language models fall far behind human performance when asked to select the correct spatial relation between two objects in an image (real examples shown).

## 1 Introduction

Pre-trained vision-language models perform well on complex tasks such as VQAv2 (Goyal et al., 2016) and Nocaps (Agrawal et al., 2019), even in the zero-shot setting (Li et al., 2023). However, recent work has re-surfaced a concern that has long plagued vision-language models (Yatskar et al., 2016; Johnson et al., 2017): new multimodal models *still* exhibit poor behavior on simple tasks like attribute attachment, counting, etc. (Yamada et al., 2022; Thrush et al., 2022; Yuksekgonul et al., 2023; Parcalabescu et al., 2021). Despite improvements, models still fail to reliably capture even

basic spatial factors of images, a prerequisite for more precise and complex reasoning benchmarks.

*But why?* In this work, we study vision-language models' performance on basic spatial relations, such as "left of" and "right of". Existing benchmarks which aim to operationalize spatial understanding such as VQAv2 and GQA (Hudson and Manning, 2019) often conflate the evaluation of spatial reasoning with other types of reasoning, such as in the GQA question "Is there a woman to the left of the person that is wearing a wetsuit?".

Hence, we first curate COCO-spatial and GQA-spatial based on the COCO (Lin et al., 2014) and GQA datasets respectively, to isolate and assess more strictly only basic spatial relations. In addition, we collect a third evaluation corpus, What′sUp, with even tighter controls. The images within COCO and GQA often contain many objects/relations, and exhibit biases that reflect our usual world (e.g., a mug is usually on a table, not under it). We manually capture controlled photographs of household objects in various positions: e.g., to overcome the social bias of dogs being photographed under tables, we (carefully, gently, and with many treats) placed a dog *on* a table and took a picture of her (see Figure 1). What′sUp consists of 205 sets of four images each, resulting in 820 images in total. Each set of images varies the underlying preposition that describes the relationship between two objects, e.g., one set of images contains a mug on, under, left of, and right of a table. Furthermore, background objects are minimized, so there is no ambiguity.

For all three datasets, our setup is as follows: for a given image, the model is given a correct caption and 1 or 3 distractor captions, which differ only by a preposition: it must select the correct one. We evaluate 18 popular vision-language models, covering various architectures (e.g., one-stack vs. two-stack), training objectives (e.g., generative vs. contrastive models), and training data. All models perform poorly across benchmarks, with many performing just a few points above random chance and all models falling far behind human performance.

Next, we investigate why these models fail to learn much about spatial relationships. All models we consider are pre-trained on large-scale image-caption corpora. We perform a corpus study of the LAION-2B dataset (Schuhmann et al., 2022), which was used to train OpenCLIP (Ilharco et al., 2021). We see that (1) common spatial prepositions occur in less than 0.2% of the training data; (2) when they do occur, they can be ambiguous or extraneous to the image, e.g., "left" defined from the viewer's perspective vs the subject's; and (3) they can often be guessed without looking at the image, e.g., "a house above water".

We consider several modeling improvements based on these findings, including: (1) renormalizing model probabilities to account for the implicit text-only prior of captions in LAION-2B; (2) replacing the preposition "behind" with one

more frequent in the training data, "in the background", as a case study to investigate if models may indeed "understand" spatial relationships (but are not surfacing that knowledge due to distribution mismatches); and (3) finetuning on several different relevant training sets (e.g., COCO-spatial/GQA-spatial training sets, preposition-containing subsets of LAION-2B, and auto-generated hard negatives with switched prepositions). None of these approaches dramatically improves model performance on understanding spatial relations.

In summary, our contributions are: (1) three new benchmarks evaluating spatial relations in vision-language models, alongside results of 18 VL models on them; (2) a study of the training data of some of these models, with observations that could explain poor model performance on the benchmarks; and (3) a study of various methods to improve model performance, with insights that could guide future research in overcoming this issue. We release code and data to encourage the same at `https://github.com/amitakamath/whatsup_vlms`.

## 2 Benchmarks

Existing benchmarks include spatial reasoning questions, such as VQAv2 (Goyal et al., 2016) and GQA (Hudson and Manning, 2019). However, instances in these corpora often conflate several types of reasoning: in GQA, over 92% of the validation questions do so. For example, the GQA question "Are there men to the left of the person that is holding the umbrella?" conflates evaluation of spatial reasoning, object relationships, and object detection – in contrast, our questions require *only* spatial reasoning about one or two objects.

Our three new evaluation corpora are presented in the same format: an image paired with several captions which differ only by a preposition. What′sUp consists of tightly controlled photographs we captured ourselves, whereas COCO-spatial and GQA-spatial are curated from well-recognized image datasets. One key contribution is that all instances in all of our corpora require *only* spatial reasoning about one or two objects, e.g., in What′sUp, we circumvent the part-and-whole problem discussed in Yamada et al. (2022) by careful construction.

Figure 2 contains examples of images from each of our three benchmarks, along with the caption options each image is paired with.

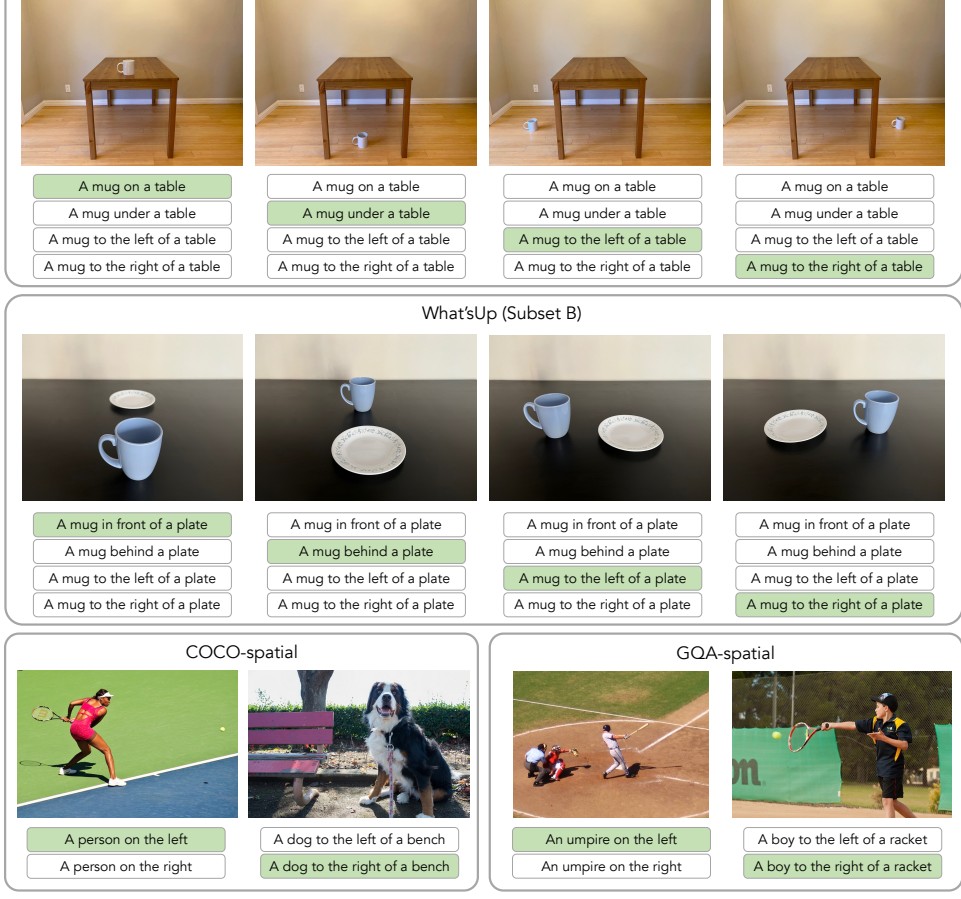

Figure 2: Examples from our three proposed benchmarks. Each image is paired with four text options in What'sUp and two text options in COCO-spatial and GQA-spatial. Given a single image and the corresponding text options, a VL model must select the correct option.

## 2.1 Collection and statistics

**What'sUp**  We captured 820 images of pairs of household objects in unambiguous spatial relation to each other. 408 of these (Subset A) contain an object on, under, left of, or right of a table, chair or armchair. The other 412 (Subset B) contain an object in front of, behind, left of or right of another object on a black tabletop. For a given object pair, each preposition is represented; thus each subset of What'sUp has equal representation of each preposition.

These images were captured with a tripod, with minimal changes between images in terms of position and lighting, except for the placement of the objects. This allows the benefit of real-world images, while exhibiting the controlled nature of synthetic images. This control has several advantages: (1) we are able to evaluate model performance on pairs or sets of images, as described in §2.2; (2) we overcome textual biases that could falsely improve

model performance, e.g. always guessing that the mug is *on* the table based on training priors; and (3) we are able to run specialized experiments studying model representations such as in §2.4.

The primary differences between the two subsets are: (1) in Subset B, the two objects are closer in size than in Subset A; and (2) in Subset B, there is no obvious prior on the spatial relationship between the two objects, whereas in Subset A, e.g., a mug would usually go *on* a table.

**COCO-spatial**  We created a benchmark from the validation set of COCO (Lin et al., 2014) using detection annotation data. We select images with only one instance of each object mentioned in the text input, where the area of each is at least 3% the area of the image. Unlike in What'sUp, these images contain objects that may embody multiple spatial relations, e.g., an object that is both to the top of and to the left of another object. Thus, we provide only caption options that are mutually ex-

clusive (to the left of vs to the right of, above vs below). Similarly for one-object images, we only test for mutually exclusive spatial relations (on the left vs on the right, on the top vs on the bottom). This benchmark contains 2687 images, with two caption options each.

**GQA-spatial** We isolated questions targeting basic spatial relations from the GQA validation dataset (Hudson and Manning, 2019), which is sourced from Visual Genome (Krishna et al., 2016). The questions we isolate are of the form "Is the *object* on the *preposition* of the image?" or "Is the *object*$_1$ to the *preposition* of *object*$_2$?", when the object(s) mentioned are all present in the image, to avoid conflation with object detection. We retain attribute-object pairs (e.g., "white car") only if the attribute does not affect the answer (e.g., there is only one car in the image), to avoid conflation with attribute detection. Similar to COCO-spatial, we select images where the area of each object in the question is at least 3% of the image. We manually filtered out noisy images, e.g., those with multiple instances of objects in the question with different spatial relations. Finally, we convert these questions to a templated caption format. This benchmark contains 1451 images, with two caption options each, due to the same ambiguity as in COCO-spatial of objects having multiple spatial relations.

## 2.2 Evaluation

**Task.** For all three benchmarks, the input is an image paired with several caption options that differ only by the preposition they contain. The model must select the caption with the correct preposition. As shown in Figure 2, for What'sUp, there are four caption options; for COCO-spatial and GQA-spatial, there are two.

**Metric.** The primary metric we use is the percentage of images for which the image-text matching score is highest for the correct caption compared to the incorrect caption(s). The controlled and balanced structure of What'sUp enables two additional metrics for that corpus: pair-wise and set-wise accuracy. Pair-wise accuracy is the accuracy on pairs of images that contain opposing prepositions. For example, if the model guesses correctly for "mug on table" *and* "mug under table", it gets one point. Set-wise accuracy is similar, but is awarded only when all four prepositions for a given object pair are guessed correctly.

**Human estimated performance.** We also estimate human performance on our three benchmarks. We sample 100 data points from each benchmark and, to ensure quality of the annotations, invite experts to voluntarily annotate the data. The annotators have all taken at least one graduate course in NLP. They are asked to determine whether the correct caption is an obvious choice, or if there is any scope for ambiguity. This estimate of human performance is 97.3% on COCO-spatial, 99% on GQA-spatial, and 100% on What'sUp.

**Models.** The models we study in the zero-shot setting are: CLIP (Radford et al., 2021) ViT-B/32 and ViT-L/14; a version of CLIP ViT-B/32 that has been finetuned on word order shuffling data called negCLIP (Yuksekgonul et al., 2023); a version of CLIP ViT-B/32 that has been initialized with RoBERTa-pretrained weights (Ilharco et al., 2021); CoCa, a model trained with generative and contrastive objectives (Yu et al., 2022); XVLM (Zeng et al., 2022) with 4M and 16M parameters; BLIP (Li et al., 2022) with 14M and 129M parameters; BLIP2 (Li et al., 2023) image-text matching head (ITM) and image-text contrastive learning head (ITC); and FLAVA (Singh et al., 2022). These models span various modeling choices: one- and two-stack models, generative and contrastive training objectives, different training data, etc.

We also study several models that have been finetuned on downstream tasks: CoCa which has been finetuned on COCO captioning; two versions of XVLM-16M that have been respectively finetuned on Flickr30K retrieval and COCO retrieval; and three versions of BLIP-14M that have been respectively finetuned on Flickr30K retrieval, COCO retrieval, and VQAv2.

Almost all of these models are capable of yielding a score representing how well a given caption matches a given image. We use this score to evaluate whether the model "selects" the correct caption from the given options for an image. As BLIP-VQA and BLIP2-ITC have a text generation head rather than a scoring head, we phrase the input as a set of questions, e.g. "Is the mug on the table?", "Is the mug under the table?", etc, and evaluate the model by measuring the probability of the responses "yes" and "no": if the probability of "yes" is highest for the gold option (or "no" is lowest for the gold option if all option responses are "no"), we award a point.

| Model | Whats-Up | COCO-spatial | GQA-spatial | Avg |
|---|---|---|---|---|
| CLIP ViT-B/32 | 31.0 | 47.4 | 46.9 | 41.8 |
| CLIP ViT-L/14 | 26.1 | 49.5 | 47.3 | 41.0 |
| NegCLIP | 34.4 | 46.9 | 46.0 | 42.4 |
| RoBERTaCLIP | 25.1 | 50.0 | 49.8 | 41.6 |
| CoCa | 29.4 | 46.7 | 47.1 | 41.0 |
| XVLM 4M | 31.5 | 61.7 | **58.7** | 50.6 |
| XVLM 16M | **41.9** | **65.0** | 58.2 | **55.0** |
| BLIP 14M | 38.5 | 54.0 | 49.8 | 47.5 |
| BLIP 129M | 30.4 | 49.3 | 49.0 | 42.9 |
| BLIP2-ITM | 37.6 | 53.0 | 49.8 | 46.8 |
| BLIP2-ITC | 29.0 | 53.7 | 51.0 | 44.6 |
| FLAVA | 30.5 | 52.6 | 51.7 | 44.9 |
| CoCa-Caption | 24.1 | 48.6 | 49.5 | 40.8 |
| XVLM-Flickr30K | 44.3 | 65.2 | 61.4 | 56.9 |
| XVLM-COCO | 42.1 | **71.0** | **68.1** | **60.4** |
| BLIP-Flickr30K | 33.8 | 54.2 | 48.9 | 45.6 |
| BLIP-COCO | 32.8 | 51.4 | 51.4 | 45.2 |
| BLIP-VQA | **47.8** | 62.0 | 58.4 | 56.0 |
| Random / Text-only | 25.0 | 50.0 | 50.0 | 41.7 |
| Human Estimate | 100.0 | 97.3 | 99.0 | 98.8 |

Table 1: Results of varied VL models on our benchmarks: models in the first section are evaluated zero-shot, and models in the second section have been fine-tuned on some downstream task: COCO captioning, retrieval on Flickr30K or COCO, or VQA. All models perform poorly on basic spatial relations.

## 2.3 Results

The performance of the models on our benchmarks is listed in Table 1. All models fall far behind human-estimated performance, with many models scoring within a few points of random chance. The number of models we evaluate allows us to draw inferences about various aspects of model design and training, as discussed below.

**Model architecture.** XVLM and BLIP2 perform better than other models in the zero-shot setting, hinting that the increased expressiveness of one-stack, cross-attention models vs the two-stack models may indeed matter in this case.

**Model size in parameters.** Scaling up model size does not necessarily improve spatial reasoning capabilities. In the case of XVLM, the 16M model outperforms the 4M model; however, CLIP ViT-B/32 outperforms CLIP ViT-L/14 and BLIP 14M outperforms BLIP 129M averaged across our three benchmarks.

**Training objective.** Despite helping on other zero-shot tasks such as ImageNet-1K (Deng et al., 2009; Yu et al., 2022), the generative training objective does not seem to encourage spatial reasoning

abilities more than a contrastive objective: CoCa scores less than CLIP ViT-B/32, and BLIP2-ITC scores less than BLIP2-ITM.

**Supervision.** XVLM is the highest-performing model of those we evaluate, likely due to its more fine-grained supervision at the bounding-box level in addition to the image-level.

**Finetuning.** Finetuning on downstream tasks appears to improve model performance sometimes, e.g. BLIP-VQA outperforms BLIP significantly, but not always, e.g. CoCa-Captioning underperforms CoCa.

**Pair/Set and One-object/Two-object accuracy.** Detailed results including pair and set accuracy for What'sUp, and one- and two-object accuracy for COCO-spatial and GQA-spatial are presented in Appendix Table 3. All models show very poor pair and set accuracy, showing their lack of understanding of the *concept* of each preposition. There does not seem to be a uniform trend of model performance on one-object images vs two-object images.

Inspection of the failure cases shows some models always predicting 1-2 prepositions for all inputs, and others predicting seemingly randomly. Overall, our data allows a very precise evaluation of spatial reasoning, revealing that these models exhibit a failure to understand basic spatial relations, despite nearing human performance on VQAv2, as in the case of BLIP-VQA.

## 2.4 Visual analogies

Next, we study the representations of CLIP models on the What'sUp Benchmark. The models *are* able to get some examples correct (e.g. "dog on a table", "dog under a table"), but as they are not able to get higher performance, particularly on the pair and set metrics, it hints that they are not learning the generalizable *concept* of "under" or other spatial relations. To study whether the representations encode these concepts in a generalizable manner, we study whether the image representations of these images exhibit the same linear analogies as studied in NLP ($king - man + woman = queen$) (Mikolov et al., 2013). We study only CLIP variants in this setting, as they alone of the models we study are trained in a manner to encourage linear recoverability. Specifically, we evaluate CLIP ViT-B/32, ViT-L/14, NegCLIP and RoBERTaCLIP.

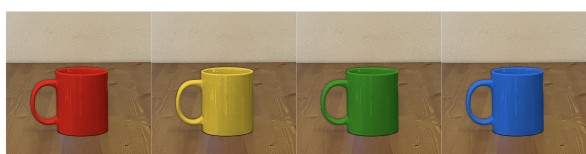

Figure 3: Example of edited images with four colors.

**Prepositions.** We select 25 sets of 4 from What'sUp Subset A: specifically, images where objects are placed around a table. We now evaluate whether $I(\text{mug on table}) - I(\text{mug under table}) + I(\text{bowl under table})$ is the closest to $I(\text{bowl on table})$, compared to $I(\text{bowl left/right/under table})$, where $I(\cdot)$ is the image representation. Given 25 objects and 4 preposition options, there are 7200 such analogies. We measure the percentage of these where our condition holds. On average, the four CLIP-based models we study achieve an analogy accuracy of only 9%. The average performance of the models when directly evaluated on the images according to our usual accuracy metric is 31%.

**Colors.** As a control test for our setup, we next study whether these linear analogies appear in the representation of various colors, which CLIP has been shown to generalize to very well (e.g., correctly identifying a blue cow). We isolate 25 objects from the What'sUp Benchmark, and edit the images to attribute one of four different colors to the object: red, yellow, green or blue, as in Figure 3. We now evaluate whether $I(\text{red mug}) - I(\text{yellow mug}) + I(\text{yellow bowl})$ is the closest to $I(\text{red bowl})$, compared to $I(\text{yellow/green/blue bowl})$, where $I(\cdot)$ is the image representation. Here, again, we have 7200 analogies and measure the percentage of times the condition holds. On average, the four CLIP-based models we study achieve an accuracy of 61%[1] – much higher than for prepositions. They also achieve 100% accuracy when directly evaluated on the color options in the same format as our basic evaluation (given one image and four caption options with different colors, select the correct caption). These experiments suggest that models appear to learn the concept of color attachments more effectively than spatial relations.

---

[1]The linear analogy accuracy is not very high, but this is perhaps not too surprising given that even performing JPEG compression before encoding changes the image representation significantly for CLIP (see https://github.com/allenai/mmc4/issues/12).

## 3 Why do they struggle? Studying LAION

All models we consider in Section 2.2 utilize large-scale image-caption corpora for pretraining. Here, we investigate one popular such corpus, LAION-2B (Schuhmann et al., 2022), to better understand why spatial relations might not be learned by models when trained on this type of data. LAION was also used to train OpenCLIP (Ilharco et al., 2021).

**Prepositions occur rarely.** We find that captions in the corpus contain common spatially specific prepositions like "under" or "left of" only 0.2% of the time (we additionally filter spatial prepositions that are used in non-spatial contexts, e.g., "under $25"). The individual frequency of each preposition is given in Appendix Table 4.

There are several reasons why this may be the case: alt-text authors may choose not to specify prepositions they feel are obvious (e.g., a house "above" the water) or ambiguous (e.g., "left" from the viewer's perspective, or from the subject of the image's perspective?); the preposition may not be important in the writer's eyes when trying to capture holistic information about the entire image in a short caption (e.g., "a cluttered kitchen", rather than "a fork to the left of a knife on a kitchen counter"); the writer may choose more casual language (e.g., "next to" rather than "to the left of"). See Berg et al. (2012) for a discussion of how descriptions manifest according to similar factors in crowdsourced image captioning corpora.

**Prepositions can be ambiguous.** Of the spatial prepositions that do occur in LAION, examination of the associated images reveals ambiguity. For example, the frame of reference could be defined from the perspective of the viewer of the photo, or of the subject of the photo — in our benchmarks, we follow the same convention as CLEVR (Johnson et al., 2017), i.e., the perspective of the viewer; however, image-text pairs in LAION are scraped from the internet, and thus follow no single convention. As another example, "in front of" could mean closer to the viewer of the photo, or ahead of a subject that is facing in a certain direction in the photo. Even the same preposition with the same meaning could have very different visual appearances, e.g. "a ball under the desk" vs "a ball under the water". A few examples are discussed in Figure 4.

| Image and caption | Discussion |
|---|---|
|  Really pleased with this startrail. Only managing approx 5hrs of darkness because of the long days. Taken between 1030pm and sunrise following day. May 31 2009 in Sth Leics, UK. Love the opposite curvature of the trails **above** and **below** the celestial equator. Olympus E3, 7-14mm lens. Just over 1000 exposures stacked in startrails. | The celestial equator is not obvious in this image, and thus the description of trails above and below it does not provide much information. |
|  Maury Determined That Was a Lie you said the next bus/train was coming up right **behind** you the half an hour wait determined that was a lie , made with livememe meme creator | The caption is a transcription of the text overlaid on the image; the image does not contain a bus or train at all. |
|  Learning objects. Fabric with sewing item and accesories which are required to learn to sew on wooden table background. Directly **above** and copy space. | Unclear what the preposition refers to. |

Figure 4: Examples of ambiguity in spatial prepositions used in LAION captions, alongside discussions thereof.

**Prepositions are rarely needed to satisfy the contrastive learning objective.** CLIP and similar models trained contrastively rely on a large batch size to obtain negative examples that require more precise visual representations. For example, the model learns a visual representation of "Bernese Mountain Dog" rather than just "dog", as there could be several types of dogs in the 32K batch. However, this is not the case for prepositions. Given the combinatorial space of all possible sentences, it is unlikely that the exact same description would apply to two images in a batch with the exception of a specific preposition. Furthermore, some preposition-object combinations are much more common, e.g., "dog under table" vs. "dog on table". Thus, we hypothesize that the model can perform well on the contrastive training objective despite ignoring spatial relationships between objects in the image.

## 4 Data-informed attempts at improvement

In this section, we operationalize our hypotheses detailed above to yield potential solutions to models' struggle with learning spatial relations.

### 4.1 Incorporating Caption Priors

The first method we consider is a re-normalization of probabilities. Intuitively, some captions are more likely on average across all images. We estimate the prior for a caption by calculating its average dot product with a large set of images from a different source to avoid test set contamination (e.g. COCO to estimate priors of a VG caption). We then use that prior to re-normalize the caption probability for a given image. Specifically, we compute a re-normalized caption probability as the difference between the un-normalized probability and the caption's calculated prior. This process is similar to the text-only normalization of Holtzman et al. (2021). This normalization encodes that $P(caption|image)$ should not depend on $P(caption)$.

Tables 5 and 6 in the Appendix contain the results of models with and without considering caption priors from different datasets. Overall, it seems that normalizing by caption priors does not tend to improve performance on What'sUp much (although a slight improvement is observed in pair and set accuracies). The priors are slightly helpful for performance on COCO-spatial and GQA-spatial, likely because those two image distributions are closer to each other than either is to What'sUp. However, overall, this approach did not drastically improve model performance on any of the benchmarks. Thus, poor performance of vision-language models cannot be attributed entirely to difficult-to-overcome text-only priors on correct options of the captions we evaluate.

### 4.2 Better prompts: don't fall (for) "behind"

From our study of the LAION-2B dataset, we see one word that is not a basic spatial preposition, but gives information about spatial relations, and has relatively high prevalence in the data: "background". This word alone appears in 0.84% of the captions, four times more than all of the other prepositions we study combined. Many of these captions describe synthetic images (e.g., "the words happy new year on a red background"), but others provide spatial information (e.g., "two people talking with some flowers in the background"). The most similar preposition we evaluate is "behind", in What'sUp Subset B.

To determine whether models understand the concept of "behind" (but this knowledge may not be accessible by using that particular word), we

do a case study of whether models trained on LAION perform better when given a prompt of "background" or "behind". We take the "in front of" and "behind" images from What′sUp Subset B (disregarding the "left of" and "right of" images), changing the text input options to (1) "$object_1$ behind $object_2$" and "$object_2$ behind $object_1$", or (2) "$object_2$ with $object_1$ in the background" and "$object_1$ with $object_2$ in the background". This allows us to evaluate only performance on "behind" vs "background" without conflating other factors such as performance on other prepositions. For CLIP ViT-B/32 and CLIP ViT-L/14 (both Open-CLIP versions trained on LAION), performance on (1) is an average of 52%, just two points above random chance, whereas performance on (2) is an average of 67%.

**Discussion.** This is a significant jump, and shows that spatial information may indeed be present in these models, but may have to be teased out more carefully. A strong caveat to these results is that the word "background" seems to be a special case: we are able to run this experiment because it appears very frequently in LAION, but we did not come across any other such words that appear frequently and provide spatial understanding. Thus, while this is an interesting thought experiment and provides hope that with more data, the issue can be mitigated, we do not believe it is the solution for models' poor performance on all spatial reasoning tasks.

### 4.3 Finetuning

Finally, we run several experiments with finetuning. Ideally, models should be able to understand basic spatial relations without finetuning, especially as finetuning tends to lose some benefits from pretraining and is tedious and expensive to do for various downstream tasks. However, we experiment with some finetuning settings with CLIP ViT-B/32 to determine whether spatial reasoning can be easily learned by our models with extra training. The results are presented in Table 2.

**Finetuning on the train equivalents of COCO-spatial and GQA-spatial.** We repeat the automated process to curate spatial relations data from GQA and COCO on the training set (rather than the validation set, which was used to create the benchmarks), dropping the filter for the objects to be at least 3% the area of the image, and dropping the human quality filter. We also combine an equal weight of COCO captions, so the model does not

| Model | Whats-Up | COCO-spatial | GQA-spatial | Avg |
|---|---|---|---|---|
| CLIP ViT-B/32 | 31.0 | 47.4 | 46.9 | 41.8 |
| + train COCO-spatial and GQA-spatial | 26.7 | 63.9 | 59.5 | 50.0 |
| + LAION-4M-prep | 33.1 | 46.0 | 47.6 | 42.2 |
| + LAION-4M-prep with neg. cap. | 29.3 | 44.4 | 46.5 | 40.1 |
| Random / Text-only | 25.0 | 50.0 | 50.0 | 41.7 |
| Human Estimate | 100.0 | 97.3 | 99.0 | 98.8 |

Table 2: Results of different types of finetuning on CLIP ViT-B/32. Even with finetuning, the results do not increase by a large margin across all benchmarks.

forget standard English. This gives us 900,000 data points, which we downsample to 300,000 for compute reasons. When we finetune on this data, we see the model improves on COCO-spatial and GQA-spatial by an average of 14.6 accuracy points. But performance *drops* on What′sUp by 4.3 accuracy points. Plausible explanations include the image distributions being different, and that the What′sUp data contains unusual placements of objects. Also, even with significant supervised in-distribution data, performance on COCO-spatial and GQA-spatial still lag significantly behind human performance (by ~50 accuracy points).

**Finetuning on a subset of LAION including prepositions.** We next isolate a subset of LAION including the prepositions we evaluate across our benchmarks. After filtering noise, this subset contains 4M image-text pairs. When finetuned on this data, performance improvements are marginal. The reasons for this could be as discussed in Section 3 – prepositions in LAION are ambiguous and rarely required to identify the image, even from a large batch (we finetune with a batch size of 2048 across 4 NVIDIA RTX A6000 GPUs).

**Finetuning on LAION-4M with hard negative captions.** Taking inspiration from Yuksekgonul et al. (2023), we add hard negative captions to the LAION-4M subset we curate, by programmatically switching the preposition with its opposite. This ensures that the model is forced to distinguish between the two in order to meet the training objective. For CLIP ViT-B/32, we observe a very high training loss, suggesting that the model cannot fit this augmented corpus.[2] We additionally track how

---

[2]Across several hyperparameter settings, we consistently observed loss of 5.0, compared to the loss of 0.01 for the same configuration without the very hard negatives.

the model allocates its probability across the batch: loss on the positive caption is similar to the loss on the negative caption, which suggests that CLIP is able to narrow text options down to those two captions, but cannot consistently learn which is correct of the two. Experiments with ViT-B/32, ViT-B/16 and ViT-L/14 all show this pattern when finetuned on both 50% and 100% of the data, implying that, at least for the training regime we consider, scaling the data or model size does not help. It is likely that an inductive bias or denser supervision is needed to enable the model to learn this, as in XVLM. The train loss curves are provided in the Appendix.

## 5   Related work

Spatial reasoning has long been evaluated by vision-language benchmarks: VQAv2 (Goyal et al., 2016), GQA (Hudson and Manning, 2019), NLVR2 (Suhr et al., 2018), CLEVR (Johnson et al., 2017) and ShapeWorld (Kuhnle and Copestake, 2017) all contain questions requiring spatial reasoning. However, many of these questions conflate several types of reasoning. Performance on these benchmarks therefore masks VL models' struggle with spatial understanding specifically.

More recently, vision-language benchmarks evaluating more specific phenomena have been proposed, testing understanding of word order (Thrush et al., 2022; Yuksekgonul et al., 2023), counting (Parcalabescu et al., 2021), object-attribute association (Yamada et al., 2022), and compositionality (Kamath et al., 2023; Ma et al., 2022). Other work including VALSE (Parcalabescu et al., 2022), VSR (Liu et al., 2023), VL-Checklist (Zhao et al., 2023) and ReCLIP (Subramanian et al., 2022) evaluate spatial reasoning in isolation, as we do in our three corpora, testing VL models' ability to match an image to the more fitting of two captions where only the spatial preposition is flipped. They show that models have room for improvement in both zero-shot and finetuned settings.

However, all of the non-synthetic benchmarks above testing spatial reasoning are based on COCO (Lin et al., 2014) or Visual Genome (Krishna et al., 2016), which are sourced from Flickr. These images tend to have many objects, usually in cluttered environments, which can confuse models trained with only image-level supervision (Yamada et al., 2022). The images also reflect biases in our usual world, such as mugs usually being *on* tables and not

*under* them[3]. Models may learn these priors and attain high scores on these benchmarks without actually attending to the images (Hsieh et al., 2023) — e.g., text-only GPT-1 (Radford et al., 2018) scores 27 accuracy points above random chance on spatial reasoning questions in VALSE. In contrast, we capture sets of photographs for What'sUp which are uncluttered, unambiguous, and contain all four preposition options for any pair of objects — thus exposing any bias models may have for the "usual" relation between two objects, as well as preventing models with such a bias from leveraging it to mask their spatial understanding abilities.

Text-to-image generation has also been shown to struggle with correctly depicting spatial relations (Gokhale et al., 2022; Hu et al., 2023). Our work sheds light on why this could be the case: e.g., DALL-E 2 (Ramesh et al., 2022) uses a frozen CLIP backbone, and as we show in our work, CLIP itself struggles with spatial reasoning.

## 6   Conclusion

In this work, we propose three new benchmarks: What'sUp, COCO-spatial and GQA-spatial, to evaluate VL models on basic spatial relations in a range of environments, with the controlled nature of What'sUp allowing us to evaluate pairs and sets of prepositions for a given object pair. We observe that all 18 models we evaluate perform poorly on these benchmarks in a zero-shot fashion. Next, we study the LAION dataset which was used to train OpenCLIP, revealing that prepositions are rare, ambiguous, and extraneous in the captions. Finally, we explore potential remedies, ultimately finding that CLIP models, at least in the regime of scale we consider, fail to even fit a large-scale training set that requires precise spatial reasoning.

How might models solve our newly proposed evaluations going forward? Three promising future directions include: (1) Auto-generation of hard negatives for spatial prepositions (and beyond) during pre-training; (2) Consideration of more expressive fine-tuned models that support image-text cross-attention and mixes of contrastive and generation objectives; and (3) Thorough scaling experiments to probe for potentially promising relationships between increasing compute of vision-language models vs. performance on our benchmarks.

---

[3]As of the time of this writing, even querying Google Image Search with "a mug under/left of/right of a table" did not yield any accurate images.

## Limitations

First, the benchmarks we propose, especially What's Up, are restricted in scale compared to benchmarks like ARO (Yuksekgonul et al., 2023) and GQA (Hudson and Manning, 2019). Second, our paper focuses on investigating how and why vision-language models struggle with basic spatial relations: our methods to improve models, while grounded in observations from our investigation, do not improve model performance significantly on all of our benchmarks. Third, our work is restricted to spatial reasoning. It would be interesting to perform a wide-scale study tackling several types of reasoning.

## Acknowledgements

We thank John Hewitt, Akhila Yerukola, Liunian Harold Li, and the anonymous reviewers for helpful discussion and feedback, as well as Travis McGuire and Emily Chua for helping us take photographs of Lucy, the star of Figure 1. This work was funded by the Allen Institute for AI. AK was additionally supported by the UCLA Computer Science Department First-Year Fellowship. KC was supported in part by DARPA MCS under contract number N660011924032, ONR N00014-23-1-2780, and a Sloan Fellowship. The views and conclusions contained herein are those of the authors and should not be interpreted as necessarily representing the official policies, either expressed or implied, of DARPA, or the U.S. Government.

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

## A   Appendix

This section contains additional results. Table 3 contains detailed results of VL models on our three proposed benchmarks. Table 4 breaks down the prevalence of various prepositions in the LAION-2B dataset, before and after removing noisy prepositions such as "under $25" — to emphasize that a direct count of word occurrence is not sufficient to understand the low prevalence of spatial relations in LAION captions. Tables 5 and 6 contain results of the experiments targeting re-normalization of caption priors. Table 7 contains detailed results of different types of finetuning on our three benchmarks. Figures 5 and 6 contain loss curves from finetuning with and without hard negative captions targeting prepositions — the train loss from the latter is about 500x smaller than the former, and the loss on the gold caption and hard negative caption is about the same, showing that the model struggles to disambiguate between the correct caption and the hard distractor, amongst the entire batch.

| | What'sUp Subset A | | | What'sUp Subset B | | | COCO-spatial | | GQA-spatial | | Indiv. |
| | Indiv. | Pairs | Set of 4 | Indiv. | Pairs | Set of 4 | One-obj | Two-obj | One-obj | Two-obj | Average |
|---|---|---|---|---|---|---|---|---|---|---|---|
| CLIP ViT-B/32 | 30.3 | 0.5 | 0.0 | 31.6 | 1.0 | 0.0 | 43.7 | 51.1 | 46.5 | 47.4 | 41.8 |
| CLIP ViT-L/14 | 26.5 | 1.0 | 0.0 | 25.7 | 2.0 | 0.0 | 49.2 | 49.8 | 46.1 | 48.5 | 41.0 |
| NegCLIP | 32.5 | 5.3 | 0.0 | 36.3 | 2.0 | 0.0 | 47.4 | 46.4 | 45.3 | 46.7 | 42.4 |
| RoBERTaCLIP | 25.2 | 2.4 | 0.0 | 25.0 | 0.0 | 0.0 | 46.3 | 53.6 | 50.8 | 48.8 | 41.6 |
| CoCa | 29.4 | 2.4 | 0.0 | 29.4 | 3.9 | 0.0 | 48.1 | 45.2 | 45.0 | 49.1 | 41.0 |
| XVLM 4M | 40.0 | 23.3 | 0.0 | 23.0 | 2.0 | 0.0 | 58.4 | 65.0 | 62.8 | 54.6 | 50.6 |
| XVLM 16M | 50.7 | 31.1 | 1.9 | 33.1 | 10.8 | 0.0 | 65.4 | 64.5 | 63.2 | 53.3 | 55.0 |
| BLIP 14M | 38.8 | 23.8 | 0.0 | 38.2 | 5.4 | 0.0 | 54.2 | 53.9 | 49.1 | 50.5 | 47.5 |
| BLIP 129M | 30.3 | 4.9 | 1.0 | 30.4 | 3.9 | 0.0 | 44.8 | 53.9 | 50.5 | 47.4 | 42.9 |
| BLIP2-ITM | 44.9 | 24.3 | 0.0 | 30.4 | 2.0 | 0.0 | 48.3 | 57.7 | 46.0 | 53.6 | 46.8 |
| BLIP2-ITC | 35.9 | 3.4 | 0.0 | 22.1 | 0.0 | 0.0 | 55.6 | 51.8 | 52.6 | 49.5 | 44.6 |
| FLAVA | 33.7 | 17.5 | 0.0 | 27.2 | 4.4 | 0.0 | 50.3 | 55.0 | 52.2 | 51.2 | 44.9 |
| CoCa-Caption | 25.5 | 1.9 | 0.0 | 22.8 | 0.0 | 0.0 | 45.9 | 51.4 | 48.5 | 50.5 | 40.8 |
| XVLM-Flickr30K | 45.1 | 16.5 | 0.0 | 43.4 | 17.2 | 1.0 | 63.1 | 67.3 | 64.7 | 58.1 | 56.9 |
| XVLM-COCO | 41.7 | 17.0 | 1.9 | 42.4 | 15.7 | 2.9 | 68.4 | 73.6 | 69.1 | 67.0 | 60.4 |
| BLIP-Flickr30K | 29.6 | 3.9 | 0.0 | 38.0 | 10.3 | 0.0 | 50.0 | 58.4 | 50.3 | 47.4 | 45.6 |
| BLIP-COCO | 35.7 | 1.9 | 0.0 | 29.9 | 2.0 | 0.0 | 46.4 | 56.4 | 50.3 | 52.6 | 45.2 |
| BLIP-VQA | 57.8 | 44.2 | 1.9 | 37.7 | 21.1 | 0.0 | 63.6 | 60.5 | 63.8 | 52.9 | 56.0 |
| Random chance | 25.0 | 6.3 | 0.4 | 25.0 | 6.3 | 0.4 | 50.0 | 50.0 | 50.0 | 50.0 | 41.7 |

Table 3: Detailed results of varied vision-language models on our benchmarks: models in the first section are evaluated zero-shot, and models in the second section have been finetuned on a downstream task: COCO captioning, retrieval on Flickr30K or COCO, or VQAv2. All models perform poorly on basic spatial relations, especially under the pair and set metrics (not included in the individual averages column).

| Preposition | % before removing noise | % after removing noise |
|---|---|---|
| in front of | 0.1084 | 0.0862 |
| behind | 0.0983 | 0.0489 |
| above | 0.0898 | 0.0422 |
| on top of | 0.0183 | 0.0134 |
| under | 0.2700 | 0.0097 |
| at the top | 0.0074 | 0.0050 |
| below | 0.0309 | 0.0040 |
| on the left | 0.0059 | 0.0038 |
| on the right | 0.0065 | 0.0028 |
| at the bottom | 0.0037 | 0.0023 |
| to the right of | 0.0011 | 0.0005 |
| to the left of | 0.0009 | 0.0005 |
| Total | 0.6412 | 0.2191 |

Table 4: Frequency of appearance of various prepositions in LAION-2B (english). The spatial relations we study represent less than 0.22% of the training data when combined, after removing noise.

| | What'sUp | | COCO-spatial | | GQA-spatial | | Average | |
|---|---|---|---|---|---|---|---|---|
| | w.o. priors | with priors | w.o. priors | with priors | w.o. priors | with priors | w.o. priors | with priors |
| CLIP ViT-B/32 | 31.0 | 30.0 | 47.4 | 54.0 | 46.9 | 46.2 | 41.8 | 43.4 |
| CLIP ViT-L/14 | 26.1 | 28.2 | 49.5 | 51.5 | 47.3 | 46.8 | 41.0 | 42.2 |
| NegCLIP | 34.4 | 32.9 | 46.9 | 51.0 | 46.0 | 47.0 | 42.4 | 43.6 |
| RoBERTaCLIP | 25.1 | 25.7 | 50.0 | 50.7 | 49.8 | 51.3 | 41.6 | 42.6 |
| CoCa | 29.4 | 32.3 | 46.7 | 49.2 | 47.1 | 47.7 | 41.0 | 43.1 |
| CoCa-Caption | 24.1 | 26.5 | 48.6 | 48.9 | 49.5 | 48.6 | 40.8 | 41.3 |
| Random chance | 25.0 | 25.0 | 50.0 | 50.0 | 50.0 | 50.0 | 41.7 | 41.7 |

Table 5: Summarized results of the experiments incorporating caption priors. For each model we have shown the best performance from different methods of calculating caption priors. Incorporating the low caption priors improves performance in some cases, but not by a large margin overall – in many cases, even with improvement the model still performs below random chance. Detailed results are shown in Table 6.

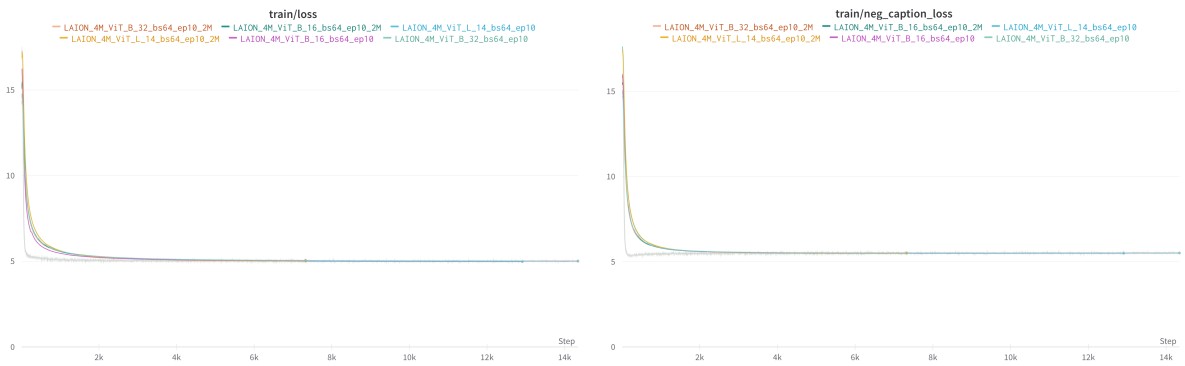

Figure 5: Train loss (left) and negative caption loss (right) when finetuning variants of CLIP on LAION-4M-prep *with* hard negatives targeting prepositions, on either the full dataset or half of the dataset (suffix _2M).

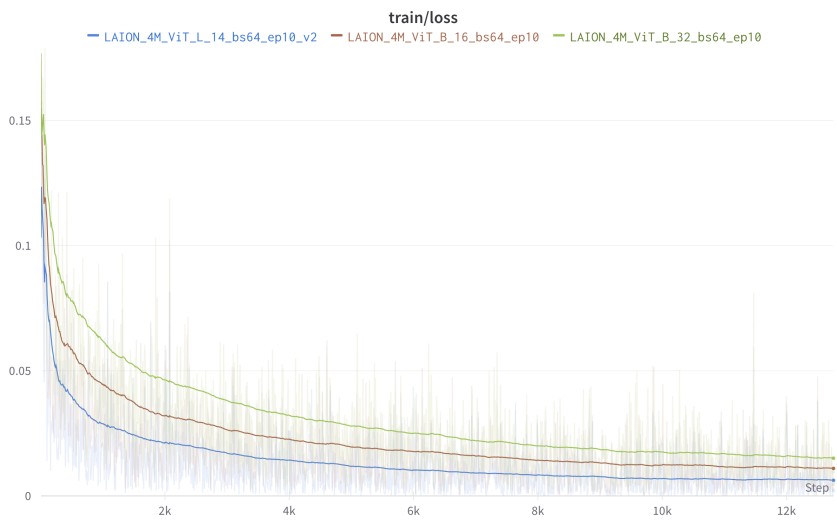

Figure 6: Train loss when finetuning variants of CLIP on LAION-4M-prep *without* hard negatives, on either the full dataset or half of the dataset. The loss is about 500x lower than in Figure 5.

**No Priors**

| | What'sUp Subset A | | | What'sUp Subset B | | | COCO-spatial | | GQA-spatial | | Indiv. Avg. w.o. COCO-spatial | Indiv. Avg. w.o. GQA-spatial |
|---|---|---|---|---|---|---|---|---|---|---|---|---|
| | Indiv. | Pairs | Set of 4 | Indiv. | Pairs | Set of 4 | One-obj. | Two-obj. | One-obj. | Two-obj. | | |
| CLIP ViT-B/32 | 30.3 | 0.5 | 0.0 | 31.6 | 1.0 | 0.0 | 43.7 | 51.1 | 46.5 | 47.4 | 39.0 | 39.2 |
| CLIP ViT-L/14 | 26.5 | 1.0 | 0.0 | 25.7 | 2.0 | 0.0 | 49.2 | 49.8 | 46.1 | 48.5 | 36.7 | 37.8 |
| NegCLIP | 32.5 | 5.3 | 0.0 | 36.3 | 2.0 | 0.0 | 47.4 | 46.4 | 45.3 | 46.7 | 40.2 | 40.6 |
| RoBERTaCLIP | 25.2 | 2.4 | 0.0 | 25.0 | 0.0 | 0.0 | 46.3 | 53.6 | 50.8 | 48.8 | 37.5 | 37.5 |
| CoCa | 29.4 | 2.4 | 0.0 | 29.4 | 3.9 | 0.0 | 48.1 | 45.2 | 45.0 | 49.1 | 38.2 | 38.0 |
| CoCa-Caption | 25.5 | 1.9 | 0.0 | 22.8 | 0.0 | 0.0 | 45.9 | 51.4 | 48.5 | 50.5 | 36.8 | 36.4 |
| Random chance | 25.0 | 6.3 | 0.4 | 25.0 | 6.3 | 0.4 | 50.0 | 50.0 | 50.0 | 50.0 | 37.5 | 37.5 |

**COCO priors**

| | What'sUp Subset A | | | What'sUp Subset B | | | COCO-spatial | | GQA-spatial | | Indiv. Avg. w.o. COCO-spatial | Improvement over no prior |
|---|---|---|---|---|---|---|---|---|---|---|---|---|
| | Indiv. | Pairs | Set of 4 | Indiv. | Pairs | Set of 4 | One-obj. | Two-obj. | One-obj. | Two-obj. | | |
| CLIP ViT-B/32 | 29.4 | 8.7 | 0.0 | 30.6 | 2.5 | 0.0 | - | - | 47.5 | 45.0 | 38.1 | -0.9 |
| CLIP ViT-L/14 | 31.1 | 5.8 | 0.0 | 25.2 | 5.4 | 0.0 | - | - | 46.0 | 47.7 | 37.5 | 0.8 |
| NegCLIP | 31.1 | 7.8 | 0.0 | 34.8 | 2.0 | 0.0 | - | - | 45.9 | 48.1 | 40.0 | -0.3 |
| RoBERTaCLIP | 24.3 | 0.0 | 0.0 | 26.0 | 1.5 | 0.0 | - | - | 50.3 | 52.3 | 38.2 | 0.8 |
| CoCa | 34.2 | 6.3 | 1.0 | 30.4 | 0.5 | 0.0 | - | - | 45.4 | 50.0 | 40.0 | 1.8 |
| CoCa-Caption | 26.7 | 3.9 | 0.0 | 26.2 | 2.0 | 0.0 | - | - | 47.9 | 49.2 | 37.5 | 0.7 |
| Random chance | 25.0 | 6.3 | 0.4 | 25.0 | 6.3 | 0.4 | 50.0 | 50.0 | 50.0 | 50.0 | 37.5 | - |

**VG priors**

| | What'sUp Subset A | | | What'sUp Subset B | | | COCO-spatial | | GQA-spatial | | Indiv. Avg. w.o. GQA-spatial | Improvement over no prior |
|---|---|---|---|---|---|---|---|---|---|---|---|---|
| | Indiv. | Pairs | Set of 4 | Indiv. | Pairs | Set of 4 | One-obj. | Two-obj. | One-obj. | Two-obj. | | |
| CLIP ViT-B/32 | 29.9 | 8.7 | 0.0 | 29.2 | 2.0 | 0.0 | 51.7 | 56.3 | - | - | 41.7 | 2.5 |
| CLIP ViT-L/14 | 30.3 | 5.3 | 0.0 | 25.5 | 5.4 | 0.0 | 50.5 | 52.5 | - | - | 39.7 | 1.9 |
| NegCLIP | 31.1 | 7.8 | 0.0 | 33.8 | 2.0 | 0.0 | 50.4 | 51.5 | - | - | 41.7 | 1.1 |
| RoBERTaCLIP | 24.8 | 0.0 | 0.0 | 26.7 | 1.5 | 0.0 | 48.1 | 53.2 | - | - | 38.2 | 0.7 |
| CoCa | 33.5 | 5.3 | 0.0 | 30.4 | 1.0 | 0.0 | 50.5 | 47.8 | - | - | 40.6 | 2.5 |
| CoCa-Caption | 26.7 | 2.9 | 0.0 | 26.2 | 1.5 | 0.0 | 50.1 | 47.8 | - | - | 37.7 | 1.3 |
| Random chance | 25.0 | 6.3 | 0.4 | 25.0 | 6.3 | 0.4 | 50.0 | 50.0 | 50.0 | 50.0 | 37.5 | - |

Table 6: Detailed results of the experiments incorporating caption priors, with different methods of calculating caption priors: no priors (top), COCO priors (middle), VG priors (bottom). Incorporating the low caption priors improves performance in some cases, but not by a large margin overall.

| | What'sUp Subset A | | | What'sUp Subset B | | | COCO-spatial | | GQA-spatial | | Indiv. Avg. |
|---|---|---|---|---|---|---|---|---|---|---|---|
| | Indiv. | Pairs | Set of 4 | Indiv. | Pairs | Set of 4 | One-obj. | Two-obj. | One-obj. | Two-obj. | |
| CLIP ViT-B/32 | 30.3 | 0.5 | 0.0 | 31.6 | 1.0 | 0.0 | 43.7 | 51.1 | 46.5 | 47.4 | 41.8 |
| FT on train COCO-spatial,GQA-spatial | 28.2 | 7.3 | 0.0 | 25.2 | 2.0 | 0.0 | 67.2 | 60.7 | 64.4 | 54.6 | 50.0 |
| FT on LAION-4M-prep | 31.6 | 1.0 | 0.0 | 34.6 | 2.9 | 0.0 | 43.1 | 48.9 | 44.3 | 50.9 | 42.2 |
| FT on LAION-4M-prep + neg. cap. | 32.0 | 0.0 | 0.0 | 26.5 | 0.0 | 0.0 | 39.9 | 48.9 | 47.3 | 45.7 | 40.1 |
| Random chance | 25.0 | 6.3 | 0.4 | 25.0 | 6.3 | 0.4 | 50.0 | 50.0 | 50.0 | 50.0 | 41.7 |

Table 7: Detailed results of different types of finetuning on CLIP ViT-B/32.