# OpenReview forum: "What's "up" with vision-language models? Investigating their struggle with spatial reasoning"
_EMNLP/2023/Conference — EMNLP 2023 Main_

### Official Review · Reviewer_sZnW · 2023-07-20

**Soundness:** 4

**Excitement:**

4: Strong: This paper deepens the understanding of some phenomenon or lowers the barriers to an existing research direction.

**Missing References:**

- Kuhnle, A., Xie, H., & Copestake, A. (2018). How clever is the FiLM model, and how clever can it be. In Proceedings of the European Conference on Computer Vision (ECCV) Workshops (pp. 0-0).
- Kuhnle, A., & Copestake, A. (2017). Shapeworld-a new test methodology for multimodal language understanding. arXiv preprint arXiv:1704.04517.

And maybe work on similar and related reasoning abilities, such as assessing objects' size:
- Pezzelle, S., & Fernández, R. (2019, November). Is the Red Square Big? MALeViC: Modeling Adjectives Leveraging Visual Contexts. In Proceedings of the 2019 Conference on Empirical Methods in Natural Language Processing and the 9th International Joint Conference on Natural Language Processing (EMNLP-IJCNLP) (pp. 2865-2876).
- Pezzelle, S., & Fernández, R. (2019, November). Big generalizations with small data: Exploring the role of training samples in learning adjectives of size. In Proceedings of the Beyond Vision and LANguage: inTEgrating Real-world kNowledge (LANTERN) (pp. 18-23).

and color:
- Schüz, S., & Zarrieß, S. (2020, July). Knowledge supports visual language grounding: A case study on colour terms. In Proceedings of the 58th annual meeting of the association for computational linguistics (pp. 6536-6542).

**Paper Topic And Main Contributions:**

The paper explores whether, and to what extent, current language-and-vision models understand spatial relationships between objects depicted in images. In particular, they focus on a bunch of relationships expressed by common English prepositions -- on, behind, on the left of, etc. -- and experiment with <image, sentence> samples from 3 datasets: COCO, GQA, and RealCLEVR, a novel dataset collected by the authors to genuinely focus on object positioning while preventing models to exploit biases due to data distribution (typical positioning, etc.). The authors evaluate several pre-trained models and show that none of them is genuinely able to understand spatial prepositions. They propose a few ways to improve models, which overall do not have a great impact on the performance.

The contributions of the paper are 1) a novel language-and-vision dataset to study spatial relationships; 2) an exhaustive investigation of current L&V models on the task; 3) an analysis of the data used to train OpenCLIP with respect to spatial prepositions; 4) some directions on how to improve model's performance on dealing with spatial relationships -- however, none of them is shown to dramatically boost model behavior.

**Questions For The Authors:**

- What do the authors think can be concluded from the experiment replacing "behind" with "background"? That more data could be necessary (if not sufficient) for solving the task? Moreover, how do these results relate or compare to the findings that fine-tuning mostly doesn't help? I think both these results are very interesting, but I'm missing some real discussion on their implications.
- Not a question, but rather a suggestion for improvement: while the Visual analogies ablation is very interesting, the motivation for having this analysis should be spelled out a bit more clearly.

**Reasons To Accept:**

- The proposed dataset, though relatively small, is a very useful resource to test models' ability to deal with spatial relationships. I particularly value the controlled nature of the dataset, which prevents models from exploiting distribution/training biases. I would just recommend the authors name it differently (see my point below).
- The set of models used for the experiments is very comprehensive and includes many different types of architectures.
- The results are extremely interesting, and highlight one key limitation of current L&V models -- spatial understanding -- that the community should take very seriously.

**Reasons To Reject:**

- It is not explained how human judgments on a sample of 100 data points were collected (if any): who are the annotators, how many per sample, how were they recruited, what was their native language, how much they were paid, what were the instructions, and what it means to compute a "conservative accuracy" on the task. These are important details that should be given.
- Not a reason to reject, per se, but I don't think it is fair to name the new dataset RealCLEVR considering that it doesn't directly build on the actual CLEVR dataset (which contains images of geometrical objects with few basic features: color, texture, size, etc.). In this sense, the name is misleading, and I would recommend the authors to consider using a different one.
- The paper lacks an insightful discussion on top of the various results: zero-shot, fine-tuning, and ablations. What do the results implicate and what do they tell us about model abilities to deal with spatial prepositions and similar expressions?


**Reproducibility:**

3: Could reproduce the results with some difficulty. The settings of parameters are underspecified or subjectively determined; the training/evaluation data are not widely available.

**Reviewer Confidence:**

4: Quite sure. I tried to check the important points carefully. It's unlikely, though conceivable, that I missed something that should affect my ratings.

**Typos Grammar Style And Presentation Improvements:**

- The abstract appears to have an overall lower writing quality compared to the paper -- I suggest the authors revise it.
- Missing capitalizations in the title and (sub)section titles; unwanted period in the title
- Contractions like "doesn't" should be avoided
- The first sentence of the abstract is not fully understandable: since VQAv2 is not only about left and right, the rhetorical question in there seems out of place
- random chance > chance level
- The captions that image is paired with are: unclear sentence
- what is the ARO benchmark?

---

> ### Author Rebuttal · Authors · 2023-08-29
>
> We thank all the reviewers for their detailed and constructive reviews. Thank you to Reviewer sZnW for pointing out our dataset as a “very useful resource with a controlled nature”, our “very comprehensive experiments”, and our “extremely interesting results [...] that the community should take very seriously”.
>
> > **Clarification on Human Judgments:**
> * The 100 data points were selected randomly from each benchmark. One of the authors familiar with the task but who had not previously seen the data (their native language is English) annotated whether or not they agreed with the ground truth annotation. The author tried to give a very conservative estimate (i.e. is there any possible interpretation that could reveal ambiguity in the data?). We will repeat the human study with multiple paid annotators, and report details in the final version of the paper.
>
> > **Changing the name of the dataset:**
> * We agree that the benchmark name could falsely imply that it is based on the CLEVR dataset. We will continue to brainstorm other names --- perhaps simply “what’s up”.
>
> > **Adding further discussion on top of the various results:**
> * In an extra camera ready page, we would add analyses about various model architectures, training data and objectives, and how they correlate with performance on the data. Example discussion points we had to cut for space and would add back: 1) we were surprised that CoCa did not outperform CLIP on our benchmarks --- we had thought that the generative training objective would be helpful vs. constrastive-only; 2) we were unsurprised to see XVLM outperform other models, as it has more fine-grained supervision at the bounding-box level in addition to the image-level.
>
> > **Clarification on Behind versus Background:**
> * The “behind” vs. “background” result shows that prompts can matter, i.e., in that case, the model can do better for some spatial relations, but only if the capability is accessed with a data-informed prompt. We were hopeful that fine-tuning might alleviate the “specific prompt required”  issue, but none of the four methods we used were able to surface similar improved behavior more broadly. We will add more discussion in revision.
>
> > **Suggestion regarding Visual Analogies:**
> * Thank you for your suggestion. Our motivation was to show that although the CLIP objective tends to make information linearly recoverable (as shown by strong results of a linear probe trained on Frozen CLIP embeddings on classification tasks like ImageNet), this does not occur for prepositions, otherwise they would have exhibited the same linear analogy behavior as Word2Vec. However, colors do exhibit this, serving as a control for the experiment. We will make the motivation clearer in the paper.
>
> > **Suggested references, writing clarity:**
> * Thank you for your suggested citations. Per other reviewers’ requests as well, we will put our work in a broader context, discussing both spatial reasoning in non-VL models (as recommended by Reviewer 4kKj), and other types of reasoning in VL models, as you and Reviewer 4kKj recommend.
> * Thank you also for your suggestions with respect to the writing. We will make the points you mentioned clearer, as well as improve the overall quality of the writing in the paper. The ARO benchmark we mention in Limitations is from Yuksekgonul et al 2023, which we will specify.
>
> Thank you again for your thoughtful comments. Please let us know if you have any further questions or comments, we would be happy to follow up!

---

### Official Review · Reviewer_4kKj · 2023-08-04

**Soundness:** 4

**Excitement:**

3: Ambivalent: It has merits (e.g., it reports state-of-the-art results, the idea is nice), but there are key weaknesses (e.g., it describes incremental work), and it can significantly benefit from another round of revision. However, I won't object to accepting it if my co-reviewers champion it.

**Missing References:**

Check my detailed comments

**Paper Topic And Main Contributions:**

The paper presents an examination of spatial relations in visual language models, aiming to address this crucial aspect in a scientific manner. To achieve this, the authors introduce a novel learning resource by curating three new corpora: COCO-prep from COCO, GQA-prep from GQA, and RealCLEVR from images. These corpora enable quantification of the model's ability to comprehend basic spatial relations. The evaluation results on 18 visual language models reveal a significant lack of understanding of the task, with most models performing not significantly better than random guessing.

The study of spatial understanding in transformer-based models is a topic that has been widely explored in the literature. The authors present their findings with clarity and provide detailed experimental descriptions to support their conclusions.

One major shortcoming is the degree paper's contribution to the field. While the authors have produced valuable insights into the spatial understanding of visual language models, the lack of spatial understanding in these models is a well-studied concept. It appears that previous research in this area, such as the works by Zhang et al. (NAACL2022), Mirzaee et al. (NAACL2021), and Singh et al. (ArXiv 2022), has not been adequately acknowledged or addressed in light of their own contributions. To be considered for acceptance in EMNLP, it is crucial for the paper to present a clear distinction of its unique contributions relative to the existing body of research that has been available for more than a year.

**Questions For The Authors:**

Check my detailed comments

**Reasons To Accept:**

- Interesting problem
- Clear writing

**Reasons To Reject:**

- Lack of novelty
- Shallow literature review

**Reproducibility:**

4: Could mostly reproduce the results, but there may be some variation because of sample variance or minor variations in their interpretation of the protocol or method.

**Reviewer Confidence:**

5: Positive that my evaluation is correct. I read the paper very carefully and I am very familiar with related work.

---

> ### Author Rebuttal · Authors · 2023-08-29
>
> We thank all the reviewers for their detailed and constructive reviews. Thank you to Reviewer 4kKj for pointing out our “interesting problem”, “clear writing” and “detailed experimental descriptions”. We will certainly put our work in a broader context, as suggested by other reviewers as well: targeting both spatial reasoning in other fields (such as image-text generation, as in Gokhale et al, arxiv 2023) and other types of reasoning in vision-language (size, color, etc, as in Zhang et al NAACL 2022).
>
> > **Three suggested citations:**
>
> * Thank you for pointing these out; we will add citations and discussion in revision. However, we believe that these works do not invalidate the novelty of our contributions. Our work studies spatial reasoning in VL models (e.g. “a dog on a table”). Zhang et al (NAACL 2022) discuss reasoning in VL models, but do not address spatial reasoning. Mirzaee et al (NAACL 2021) discuss spatial reasoning, but in text-only models, not VL models. Singh et al (Arxiv 2022) discuss one very specific spatial relation in VL models: relative elevation above the ground.
> * To our knowledge, our work is the first to (1) propose spatial reasoning data in a very controlled setting – one cannot game it with text-only priors; (2) study rare spatial relations (e.g. “a dog on a table”); (3) perform a detailed study of a large number of VL models in this setting; and (4) perform comprehensive experiments investigating why the models perform poorly. We believe that our contributions and novelty still hold in light of the relevant work (including Liu et al, TACL 2023, as discussed in L611-618, and Parcalabescu et al, ACL 2022 --- please see our response to Reviewer ciVg for discussion about this paper).
>
> Thank you again for your thoughtful comments. Please let us know if you have any further questions or comments, we would be happy to follow up!

---

### Official Review · Reviewer_ciVg · 2023-08-05

**Typos Grammar Style And Presentation Improvements:** L625
**Soundness:** 4

**Excitement:**

4: Strong: This paper deepens the understanding of some phenomenon or lowers the barriers to an existing research direction.

**Missing References:**

You cite Letitia Parcalabescu, Albert Gatt, Anette Frank, and Iacer Calixto. 2020. "Seeing past words: Testing the cross-modal capabilities of pretrained v&l models on counting tasks", while their follow-up work on VALSE is even more relevant because it includes spatial relations and even more phenomena: https://aclanthology.org/2022.acl-long.567/

**Paper Topic And Main Contributions:**

This paper is a great study of how well VLMs (Vision and Language Models) capture spatial relationships. It is comprehensively covering VLMs, testing 18 of them (discriminative and generative). It also provides an analysis of the LAION dataset, exploration of different pretraining strategies, but with little success.

**Reasons To Accept:**

The contributions of this paper are substantial:

* The number of tested VL models is astounding: 18!

* The authors have carefully introduced new data (dog on the table) to also capture rare spatial relations.

* Interesting analysis of LAION-2B which is used as a pretraining corpus for VLMs. It is surprising that the strategies which come out of these analyses, do not work too well. This is really good to know.

**Reasons To Reject:**

* I have to say that even after reading this paper, I now do not feel more informed about why VLMs fail at spatial reasoning than before. It still seems that we have no idea of what goes on in these models and the combination of the training task and rarity of some prepositions in the training data are to blame, but the exploration of data-informed improvements did not succeed. Why? This makes the paper's impact onto the community less than I expected from title, abstract and introduction.

* The paper does not make very clear what previous work on spatial relations does and how it differs from it.

**Reproducibility:**

5: Could easily reproduce the results.

**Reviewer Confidence:**

4: Quite sure. I tried to check the important points carefully. It's unlikely, though conceivable, that I missed something that should affect my ratings.

---

> ### Author Rebuttal · Authors · 2023-08-29
>
> We thank all the reviewers for their detailed and constructive reviews. Thank you to Reviewer ciVg for pointing out our “substantial contributions”, “astounding number of tested models”, “carefully introduced data capturing rare spatial relations”, and “interesting analysis”.
>
> > **Why VLMs fail at spatial reasoning / Why data-informed improvement methods didn’t succeed:**
>
> * We agree that the rarity of clearly depicted spatial prepositions in the data (see Figure 4 and Table 4), along with the training objectives not incentivizing spatial reasoning, contribute to the models’ poor performance. As to why the data-informed improvements did not succeed, we find this surprising and intriguing as well, and expect that further research is necessary to answer this question.
> * The limited improvement mirrors results shown on the Visual Spatial Reasoning benchmark (Liu et al, TACL 2023), where finetuning on in-domain spatial reasoning data only boosts performance of the top-performing model (LXMERT) to about 70%, where human performance is 95%.
> * Although finetuning with hard negatives has been shown to help with word order shuffling (Yuksekgonul et al, ICLR 2023), counting (Paiss et al, arxiv 2023) and the VL-checklist benchmark (Doveh et al, CVPR 2023), we see that for prepositions, the models struggle to fit even the training data when finetuned with hard negatives (L569-585) --- the loss is split between the correct caption and the hard negative caption for the image, hinting that even an additional loss component targeting those two in the finetuning objective (as in Paiss et al 2023 and Doveh et al, 2023) would not be helpful.
> * We believe that these results show that more research is required to improve spatial reasoning across all model architectures, potentially in the form of denser supervision.
>
> > **Previous work on spatial relations:**
>
> * Thank you --- we will expand our discussion of related work, as suggested by the other reviewers as well. Although spatial reasoning has not been extensively studied in vision-language (Liu et al, TACL 2023; Parcalabescu et al, ACL 2022), it has been studied in text-to-image generation (Gokhale et al, arxiv 2023) and is also important in vision-language navigation (Shridhar et al, CVPR 2020). Per other reviewers’ comments, we also plan to add other types of reasoning in VL beyond spatial reasoning (such as color, size, etc) to the literature review: for specific citations, please see our response to the other reviewers.
>
>
> Thank you for pointing out VALSE (Parcalabescu et al, ACL 2022) --- this is relevant and interesting work; we will discuss this in the revision! We differ from this work in the controlled nature of our data, as well as the rare spatial relations we study. Consider the random sample of the spatial relations in VALSE in Table 9: GPT-4 text-only is able to resolve all of them because, unlike our work, the images depict the expected commonsense scenario (e.g. “A cow stands on a sidewalk _outside_ a building.”, “A cow stands on a sidewalk _in_ a building.”).
>
> Thank you again for your thoughtful comments. Please let us know if you have any further questions or comments, we would be happy to follow up!

---

### Meta-Review · Area_Chair_9Lht · 2023-09-20

**Recommendation:** 4

**Metareview:**

The paper investigates how Vision and Language Models (VLMs) grasp spatial relationships. It undertakes a comprehensive examination of VLMs, evaluating various models. An analysis of the LAION dataset is presented, along with the introduction of several new corpora, namely COCO-prep from COCO, GQA-prep from GQA, and RealCLEVR. Despite the meticulous experimentations and introduction of new data to understand rare spatial relations, the results indicate that most VLMs are not performing exceptionally better than random thinking in spatial reasoning. Moreover, the study supplies insights but sometimes advances our understanding of why the models fail in this domain.

**Reason To Accept**

- Contributions: The paper's contributions are notable, especially with testing a significant number of VLMs (18 in total).
- Data Introduced: The authors have integrated new data to evaluate rare spatial relations, which is commendable.
- Clarity: The findings are clearly presented, detailed experimental descriptions support their conclusions, and the writing is clear.
- Relevance: The issue of spatial understanding in models is crucial, and this paper's results shed light on existing shortcomings.

**Reason To Reject**
- Ambiguous Impact: Even after extensive exploration and analysis, the paper must significantly advance the understanding of why VLMs fail at spatial reasoning.
 - Literature Review: The paper needs to adequately differentiate its contributions from previous significant works in this area. Key works like Zhang et al. (NAACL2022), Mirzaee et al. (NAACL2021), and Singh et al. (ArXiv 2022) are insufficiently acknowledged.
- Missing Details: The paper must explain essential details about human judgments on specific data points, such as annotator information, recruitment, and calculation methods.
- Limited Insightful Discussion: The paper could benefit from a more in-depth discussion and analysis of the results, especially regarding the implications of the models' abilities concerning spatial prepositions.

**Overall**
The paper contributes valuable insights into the domain of spatial relationships in VLMs. However, the reviewers have highlighted several concerns that limit its potential impact. To be considered for a top-tier conference, it would be essential to address the gaps in a literature review, provide more in-depth discussions on the results, and clarify ambiguities around.

---

### Decision · Program_Chairs · 2023-10-07

**Decision:**

Accept-Main

**Comment:**

The paper investigates how Vision and Language Models (VLMs) grasp spatial relationships. It undertakes a comprehensive examination of VLMs, evaluating various models. An analysis of the LAION dataset is presented, along with the introduction of several new corpora, namely COCO-prep from COCO, GQA-prep from GQA, and RealCLEVR. Despite the meticulous experimentations and introduction of new data to understand rare spatial relations, the results indicate that most VLMs are not performing exceptionally better than random thinking in spatial reasoning. Moreover, the study supplies insights but sometimes advances our understanding of why the models fail in this domain.

**Reason To Accept**

- Contributions: The paper's contributions are notable, especially with testing a significant number of VLMs (18 in total).
- Data Introduced: The authors have integrated new data to evaluate rare spatial relations, which is commendable.
- Clarity: The findings are clearly presented, detailed experimental descriptions support their conclusions, and the writing is clear.
- Relevance: The issue of spatial understanding in models is crucial, and this paper's results shed light on existing shortcomings.

**Reason To Reject**
- Ambiguous Impact: Even after extensive exploration and analysis, the paper must significantly advance the understanding of why VLMs fail at spatial reasoning.
 - Literature Review: The paper needs to adequately differentiate its contributions from previous significant works in this area. Key works like Zhang et al. (NAACL2022), Mirzaee et al. (NAACL2021), and Singh et al. (ArXiv 2022) are insufficiently acknowledged.
- Missing Details: The paper must explain essential details about human judgments on specific data points, such as annotator information, recruitment, and calculation methods.
- Limited Insightful Discussion: The paper could benefit from a more in-depth discussion and analysis of the results, especially regarding the implications of the models' abilities concerning spatial prepositions.

**Overall**
The paper contributes valuable insights into the domain of spatial relationships in VLMs. However, the reviewers have highlighted several concerns that limit its potential impact. To be considered for a top-tier conference, it would be essential to address the gaps in a literature review, provide more in-depth discussions on the results, and clarify ambiguities around.